# Preparation of Mesophase Pitch with Fine-Flow Texture from Ethylene Tar/Naphthalene by Catalytic Synthesis for High-Thermal-Conductivity Carbon Fibers

**DOI:** 10.3390/polym16070970

**Published:** 2024-04-02

**Authors:** Xubin He, Xiao Wu, Kui Shi, Shipeng Zhu, Dong Huang, Hongbo Liu, Jinshui Liu

**Affiliations:** 1Hunan Province Key Laboratory for Advanced Carbon Materials and Applied Technology, College of Materials Science and Engineering, Hunan University, Changsha 410082, China; hexubin@hnu.edu.cn (X.H.); wuxiao@hnu.edu.cn (X.W.); hd52923212@hun.edu.cn (D.H.); hndxlhb@163.com (H.L.); 2Key Laboratory of Advanced Functional Composite Materials, Aerospace Research Institute of Materials and Processing Technology, Beijing 100076, China; carbonfiber703@163.com; 3Hunan Province Engineering Research Center for High Performance Pitch-Based Carbon Materials, Hunan Toyi Carbon Material Technology Co., Ltd., Changsha 410000, China; 4State Key Laboratory of Advanced Design and Manufacturing for Vehicle Body, Hunan University, Changsha 410082, China

**Keywords:** ethylene tar, mesophase pitch, catalytic synthesis, structure and properties

## Abstract

Mesophase pitch is usually prepared by radical polymerization or catalytic polymerization from coal tar, petroleum, and aromatic compounds, and the catalytic synthesis of mesophase pitch from pure aromatic compounds is more controllable in the preparation of high-quality mesophase pitch. However, the corrosive and highly toxic nature of the catalyst has limited the further development of this method. In this study, mesophase pitch was synthetized using ethylene tar and naphthalene as raw materials and boron trifluoride diethyl etherate as a catalyst. The effect of the catalytic reaction on the structure and properties of the mesophase pitch was investigated. The results show that naphthalene plays an important role in the mesophase content and reaction pressure (from above 6 MPa to 2.35 MPa). Mesophase pitch with fine-flow texture can be prepared by introducing more methylene groups, naphthenic structures, and aliphatic hydrocarbons during synthesis. Carbon fibers prepared from mesophase pitch show a split structure, and the thermal conductivity is 730 W/(m·K). This work provides theoretical support for lower toxicity and causticity and for reaction-controlled technology for the synthesis of high-purity mesophase pitch.

## 1. Introduction

Mesophase pitch (MP) with fine-flow texture has high purity, high carbon content, regular molecular arrangement, and good fluidity and is an excellent precursor for the preparation of needle coke, carbon foam, carbon fiber, etc. [1,2,3,4]. Pitch-based carbon fibers (CFs) prepared from mesophase pitch have the characteristics of light weight, high strength, high modulus, and high thermal conductivity, which make them one of the promising high performance fibers in applications for the aerospace industry, advanced industrial manufacturing, and other fields [5,6,7,8]. Mesophase pitch is usually prepared by radical polymerization [9,10,11] or catalytic polymerization [12,13,14,15,16] from coal tar, petroleum tar, and aromatic compounds. It is hard to synthesize 100% wide-area streamlined mesophase pitch due to the difficulty in controlling the radical polymerization of the coal tar and petroleum tar. In comparison, the catalytic polymerization of high-purity aromatic compounds makes the pitch molecules retain a large amount of cycloalkane structures and alkyl side chains, thereby ensuring better molecular flexibility while maintaining flatness, making it easier to form mesophase pitch with relatively centralized molecular weight, a low softening point, good solubility, and a controllable molecular structure [17]. Therefore, the catalytic synthesis of mesophase pitch from pure aromatic compounds is more controllable for the preparation of high-quality mesophase pitch. Mochida et al. [12] synthesized mesophase pitch from pure aromatic hydrocarbons with HF/BF_3_ as catalyst. The prepared mesophase pitch had excellent performance and was industrialized by the Mitsubishi Chemical Company of Japan. However, the corrosive and highly toxic nature of the HF/BF_3_ catalyst has limited the further development of this method.

Ethylene tar (ET) is a by-product of the industrial production of ethylene from light distillates of petroleum crude oil, mainly composed of various alkanes, C_8_~C_15_ aromatics, aromatic olefins, and heterocyclic compounds containing N, S, O, and other elements [18,19], and is one of the high-quality raw materials for pitch-based carbon fibers because of its wide range of sources, low content of metallic impurities, high content of aromatic hydrocarbons, etc. Mochida et al. [20] prepared mesophase pitch with high solubility, a low softening point, and a well-developed anisotropic structure by catalytic polycondensation of ethylene tar using AlCl_3_ as a catalyst. It was found that the AlCl_3_ residue was difficult to remove, which seriously reduced the mechanical strength of the carbon fibers. Therefore, the development of a novel synthesis method with lower toxicity and causticity, a controllable reaction, and low catalyst residue is of great research significance.

In this study, mesophase pitch was synthesized using ethylene tar as a raw material and boron trifluoride diethyl etherate as a catalyst. The boron trifluoride diethyl etherate possesses lower levels of toxicity and causticity, can be easily removed during heat treatment, and can be recycled and reused through condensation to minimize environmental impact. Refined naphthalene was also used as an additive in order to further adjust the molecular structure of the pitch. The influence of the synthesis process on the structure and performance of the mesophase pitch and its carbon fibers was investigated. The study provided a reduced toxicity and causticity and a reaction-controlled technology for the synthesis of high-purity mesophase pitch.

## 2. Materials and Methods

### 2.1. Materials and Equipment

The elemental composition of ethylene tar (PetroChina Dushanzi Petrochemical Company, <20 ppm) is 0.04 wt.% N, 92.15 wt.% C, 7.19 wt.% H, and 0.08 wt.% S. Refined naphthalene (Baoshan Iron and Steel Group) has a purity of 98.5% and a melting point of 80.5 °C. Boron trifluoride diethyl etherate (commercially available) contains 47 wt.% boron trifluoride.

The pitches were synthesized in a high-pressure reactor and a normal-pressure reactor. The high-pressure reactor was equipped with a pressure gauge and a 6 MPa bursting disc.

### 2.2. Preparation of Mesophase Pitch

The preparation process for the ethylene-tar-derived mesophase pitch was divided into three steps. Firstly, ethylene tar, refined naphthalene, and boron trifluoride diethyl etherate were added into the high-pressure reactor in a certain ratio. The mass ratio of raw material (refined naphthalene and ethylene tar) to catalyst (boron trifluoride ether) was 4:1, and the content of refined naphthalene was 0~50%. Then, the reactor was heated to the target temperature with an oil bath under nitrogen atmosphere for the self-pressurization reaction. At the end of the reaction, the reaction kettle was taken out and naturally cooled to room temperature. Secondly, the supernatant was removed and the bottom precipitate was placed in the normal-pressure reactor heated at 340 °C for 5 h under nitrogen atmosphere to remove the catalyst, and high-purity precursor pitch was obtained and labeled as CP-X-T. X and T represent the mass fractions of refined naphthalene and reaction temperature, respectively. Finally, the high-purity precursor pitch was heated to 380~400 °C under nitrogen atmosphere for several hours to synthesize the ethylene-tar-derived mesophase pitch. The samples were labeled as MP-X-T.

Naphthalene-derived catalytic pitch and mesophase pitch were synthesized as the control group and labeled as CP-N and MP-N, respectively. Refined naphthalene was used as a raw material, and both hydrogen fluoride and boron trifluoride ether (HF/BF_3_) were used as catalysts. The synthesis method was the same as for the ethylene-tar-derived mesophase pitch.

### 2.3. Preparation of Pitch-Based Carbon Fiber

Melt spinning of pitch was carried out with custom-made single-hole spinning equipment (the length/diameter of spinneret is 0.4 mm/0.2 mm) at 330 °C under 0.8 MPa nitrogen with a rotating speed of 300 m/min. Then, the prepared green fibers were stabilized by pre-oxidation in a 500 mL/min air atmosphere at 310 °C for 1 h with a heating rate of 0.5 °C/min and carbonized in a 200 mL/min nitrogen atmosphere at 1000 °C for 0.5 h with a heating rate of 5 °C/min to obtain the carbon fibers. Finally, graphite fibers (GFs) were obtained through further graphitization at 2800 °C and labeled as CF-X-T.

### 2.4. Characterization

The softening point (SP) was measured with a DP70 softening point apparatus (Metter Toledo, Zurich, Switzerland) in a nitrogen atmosphere with a heating rate of 3 °C/min. Anisotropic textures of the pitches were observed with a BX53 polarization microscope (Olympus, Tokyo, Japan) after polishing. Fourier-transform infrared (FT−IR) spectroscopy was obtained using the KBr disc technique in a Nicolet iS10 FTIR spectrometer (Thermo Fisher Scientific, Waltham, MA, USA) with a resolution of 4 cm^−1^. XRD analyses were performed using a D-5000 diffractometer (Siemens, Berlin, Germany) with Cu-Kα radiation (λ = 0.15406 nm) generated at 35 kV and 30 mA at a step of 0.02° for 2θ values between 10° and 90°. The interlayer spacing can be calculated from the Bragg equation, d_002_ = λ/2sinθ. The stacking height Lc can be determined using the Scherrer formula, Lc = kλ/(β_002_*cosθ), where k is the shape factor 0.89 and β_002_ represents the half-width of the 002 plane diffraction peak [21]. And crystallite diameter can be determined using the Takahashi formula, La = 0.095/(d_002_-0.3354) [22]. The graphitization degree (G) can be calculated with the following equation: G = (0.3440-d_002_)/(0.3440-0.3445). ^1^H-NMR and ^13^C-NMR were conducted with a Bruker 400 MHZ Advance Spectrometer (Bruker, Germany). The solvents used for ^1^H-NMR and ^13^C-NMR were deuterated benzene (C_6_D_6_) and deuterochloroform (CDCl_3_), respectively. The internal standard was TMS and the relaxation reagent was Cr(acac)_3_. The surface morphology and the structure of the transverse sections of the fibers were observed with a TESCAN VEGA3 field emission scanning electron microscope (TESCAN, Brno, Czech Republic) with an accelerating voltage of 20 kV. The tensile strength and tensile modulus of the GFs were determined with an XQ-1C single-filament machine (Shanghai New Fiber Instruments Co., Shanghai, China) with a gauge length of 20 mm. The electrical resistivity (ρ) was measured with a Aim-TTi BS407 micro-ohmmeter (Aim-TTi, Huntingdon, UK), and the thermal conductivity was calculated with the formula λ = 1261/ρ [23].

## 3. Results

### 3.1. Control of Reaction Pressure and Yield of Catalytic Polymerization

Table 1 shows the effect of the mass fraction of the refined naphthalene in the raw materials on the reaction pressure and yield of the pitch in the self-pressurization reaction at 220 °C. It can be seen from Table 1 that as the mass fraction of the refined naphthalene increases, the reaction pressure decreases first and then increases. It reaches the minimum when the addition amount of the naphthalene is 20~30%. The yield of the catalytic polymerization rises first and then falls, and the yield is at maximum when the addition amount of naphthalene is 20~30%, indicating that naphthalene is involved in the catalytic synthesis. Without refined naphthalene, ethylene tar directly reacts with boron trifluoride etherate, and the boron trifluoride is decomposed at high temperature and causes extreme pressure (>6 MPa), which increases the uncontrollability of the system, and the catalyst cannot participate effectively in the liquid phase reaction. When the mass fraction of refined naphthalene increases from 5% to 30%, the catalytic polymerization pressure decreases from 6 MPa to 2.3 MPa. It is possible that the added refined naphthalene and boron trifluoride ether form a liquid complex and then the alkylation reaction occurs, which improves the utilization rate of the boron trifluoride in the ethylene tar reaction system, making the boron trifluoride and aromatic hydrocarbon compound form a stable complex, thus significantly reducing the reaction pressure. However, as the mass fraction of the refined naphthalene increases from 30% to 50%, the pressure of the reaction system increases from 2.3 MPa to 3.2 MPa. It is because the excessive and unreacted naphthalene vaporizes and further increases the pressure of the system. In this experiment, the optimal mass fraction of naphthalene is 20~30%.

Figure 1 shows catalytic synthesis pressure curves versus temperature and holding time. It can be seen from Figure 1a that with the reaction temperature increasing, the reaction pressure gradually increases. The process can be divided into two stages: (1) RT~220 °C, where the reaction pressure increases slowly, and (2) 220~230 °C, where the reaction pressure increases significantly because of intense boiling of the naphthalene under high temperature. As can be seen from Figure 1b, with the extension of the holding time, the reaction pressure first rises and then falls. The process can be divided into three stages: (1) 0~60 min, where the reaction pressure increases significantly and reaches a maximum of 2.4 MPa at 60 min; (2) 60~150 min, where the reaction pressure decreases significantly (it is expected that the addition of the refined naphthalene plays a positive role in the catalyst complexation with the molecules in the system so the self-pressurization of the system is suppressed); and (3) 150~330 min, where the reaction pressure decreases slowly and stabilizes.

### 3.2. Analysis of Mesophase Conversion

The content of the mesophase has a crucial influence on the properties of the carbon materials. Pitch with high mesophase content, suitable softening point, and viscosity/temperature performance usually requires a planar molecular structure and relatively uniform molecular weight distribution in order to prepare high-quality mesophase-pitch-based carbon fibers [24]. Figure 2 shows the polarized texture images of the mesophase pitch (SP 270 ± 2.5 °C) prepared with different mass fractions of refined naphthalene at 210 °C (a–c) and 220 °C (d–g), respectively. The mesophase pitch catalytically synthesized with HF/BF_3_ was used as the control group (h).

Figure 2a–c show polarized-light micrographs of mesophase pitch whose refined naphthalene mass fraction is 20~30 wt.% and whose reaction temperature is 210 °C. It is observed that the content of the mesophase gradually increases with the mass fraction of the refined naphthalene increasing, indicating that the conversion ability of the mesophase becomes stronger. However, a certain amount of isotropic composition and mosaic structure are also observed. It can be seen from Figure 2d–f that under the reaction temperature of 220 °C, the mesophase content is about 80% in MP-20-220. The non-mesophase parts present isotropic blocks or isotropic spheres with diameters less than 10 μm. The mesophase content is about 90% in MP-25-220, and the non-mesophase part is mainly isotropic spheres with diameters less than 10 μm. When the mass fraction of the refined naphthalene is 30%, a 100% fine linear mesophase texture is formed due to airflow purge at high viscosity. Figure 2g shows the polarized structure of ethylene-tar-derived mesophase without refined naphthalene addition. The mesophase content is about 20%, and the distribution is in a long strip. The reason for the low mesophase content is that the reactivity of the catalytic polymerization is low without the naphthalene. Figure 2h shows the polarized texture of the mesophase pitch synthesized with refined naphthalene using HF/BF_3_ as catalyst. The prepared mesophase pitch exhibits a good wide-area rheological polarized structure. It can be concluded that the conversion activity of the mesophase is effectively improved by using refined naphthalene as an auxiliary agent, and 100% mesophase pitch can be obtained when the mass fraction of the refined naphthalene is 30% at 220 °C. In addition, the isotropic components in the mesophase are believed to seriously affect the spinning performance of pitch, thereby affecting the structure and performance of the subsequent carbon fiber.

### 3.3. FT−IR of Mesophase Pitches

In order to further understand the molecular structure of the ethylene-tar-derived mesophase pitch, FT−IR was used to analyze the molecular structure of different pitches in different stages.

As shown in Figure 3, the infrared spectra of the two series of catalytic pitches and mesophase pitches show strong vibrations at 750~870 cm^−1^, 1450 cm^−1^, 1600 cm^−1^, and 2700~2970 cm^−1^, which are attributed to the out-of-plane vibration of aromatic C-H, the bending vibration of methylene (including naphthenic) C-H, the stretching vibration of the C=C of the aromatic rings, and the stretching vibration of methyl C-H, respectively [25,26]. It indicates that the two series of pitches eventually form polycyclic aromatic hydrocarbon molecules rich in methyl and methylene groups (including naphthenic structures). The peaks at 3040 cm^−1^ (aromatic CH stretching vibration) and 750–870 cm^−1^ of CP-N and MP-N show stronger signals than those of CP-30-220 and MP-30-220, especially at 750 cm^−1^ (mono-substituted aromatic CH out-of-plane bending vibration), indicating that these pitches contain more aromatic hydrogens and cata-condensed structures [27]. In contrast, the structure of ethylene tar is more complicated, and the pitches are composed of a large number of aliphatic hydrocarbons and olefins. After catalytic polymerization, the out-of-plane vibrations of the aromatic rings of the ethylene-tar-derived pitch are not obvious, but the C=C double bond stretching vibrations located at 1600 cm^−1^ are more obvious compared with the naphthalene-derived pitch, indicating that the catalytic polymerization tends to be peri-condensed. Moreover, there are obvious peaks at 1020 cm^−1^ for CP-30-220 and MP-30-220, indicating that the oxygen in the ether is involved in the reaction. However, both of them contain a large number of naphthenic hydrocarbons, methylene structures (1450 cm^−1^, 2700~2970 cm^−1^), and shorter-chained aliphatic structures (1378 cm^−1^), which keep their low viscosity and subsequently form a streamlined mesophase structure.

### 3.4. ^1^H-NMR and ^13^C-NMR of Mesophase Pitch Precursors

Figure 4 shows the ^13^C-NMR and ^1^H-NMR analyses of CP-30-220 and CP-N. The integration results based on the peak-area normalization method are listed in Table 2.

CP-30-220 has a similar structure to CP-N, but the H_ar_ and C_ar_ are small, indicating a low aromaticity. Higher H_α_ and C_CH2_ indicate abundant branched-chain, methylene, or naphthenic structures. The ^13^C-NMR spectrum shows that CP-30-220 contains more branched aliphatic hydrocarbons (0~20 ppm), bridge methylene (33~45 ppm), and aromatic bridge carbons (129~137 ppm). The naphthenic structure of CP-N (20~33 ppm) is better developed [28,29]. The results are consistent with the previous FT−IR results, which indicate the catalytic polymerization of ethylene tar tends to be peri-condensed.

### 3.5. The Graphitized Structure of Mesophase Pitch

The lamellar aromatic structure of mesophase pitch plays an important role in the formation of the carbon network structure with fewer defects after heat treatment, developing a nearly perfect graphite crystal structure. MP-25-220, MP-30-220, and MP-N were directly graphitized at 3000 °C to obtain graphite powder, and the graphitized structure was analyzed by XRD with Si as the internal standard. Figure 5 shows the XRD patterns of GP-25-220, GP-30-220, and GP-N. It can be seen from Figure 5 that a sharp diffraction peak appears near 26°, which belongs to the (002) plane diffraction peak [30], indicating a perfect crystal structure after graphitization. The crystal parameters of the graphitized samples obtained by fitting the full XRD patterns are shown in Table 3.

Compared with GP-N and GP-30-220, GP-25-220 has a larger d_002_, minimum graphitization degree (91.79%), and minimum La. It indicates that the ethylene-tar-derived mesophase pitch with low mesophase content and a small amount of mosaic structures encounters more difficulty in forming graphitized crystals. It is because the aromatic nucleus plane of the mesophase pitch without 100% mesophase content encounters more difficulty in forming larger-scale perfect carbon network planes. In comparison with GP-N, the crystallite parameters of GP-30-220 are smaller but similar, indicating that ethylene tar pitch with a 100% fine streamlined mesophase structure can become perfect graphite crystallites that are the same as the wide-area streamlined naphthalene mesophase pitch after graphitization.

### 3.6. Preparation of Carbon Fibers

The spinning performance of the two ethylene-tar-derived pitches with different mesophase contents was investigated and is shown in Table 4. The mesophase contents of MP-25-220 and MP-30-220 are about 90% and 100%, respectively. The continuous spinning time of MP-25-220 was short (<10 min), and the minimum diameter of the fibers is 20~25 μm, indicating poor drawability of the pitch fibers, while MP-30-220 can be spun for a long time, and the minimum diameter is 13~16 μm.

SEM images of CF-25-220 and CF-30-220 are shown in Figure 6. It shows that no obvious interface orientation is found in CF-25-220, and there are several holes in the fiber sections. It can be explained as follows: on the one hand, the diameter of the spun fiber is large because of the pitch’s poor drawability, and thus the core of the pitch fibers is not sufficiently oxidized under the same oxidation conditions, leading to melt decomposition during subsequent heat treatment. On the other hand, the thermal stability of the isotropic component is low, and it easily decomposes during heat treatment and forms holes inside the fibers. In addition, Table 5 shows that the tensile strength, tensile modulus, and thermal conductivity of the CF-30-220 are 2.6 GPa, 620 GPa, and 730 W/m∙K, while the tensile strength, tensile modulus, and thermal conductivity of the CF-25-220 are 2.2 GPa, 850 GPa and 410 W/m∙K, respectively. It is because the CF-30-220 has a typical split-radial structure [31], indicating high orientation and perfect graphite crystallites of the carbon fibers. No structural defects, such as pores, are observed in the cross-sections because the pitch fibers composed of 100% mesophase have a small diameter and are easily oxidized.

In summary, the content and texture of the mesophase have direct influences on the structure and thermal conductivity of the carbon fibers. Mesophase pitch with 100% mesophase content can be obtained by using ethylene tar and 30 wt.% naphthalene as raw materials, which is an excellent precursor for the preparation of pitch-based graphite fibers with high thermal conductivity.

## 4. Conclusions

(1) Mesophase pitch is synthesized using ethylene tar as a raw material and boron trifluoride diethyl etherate as a catalyst, and refined naphthalene is used as an additive in order to further adjust the molecular structure. The reaction pressure is reduced from above 6 MPa to below 3 MPa. The polarizing structure of the mesophase pitch is also changed from a large number of isotropic components to a 100% anisotropic structure.

(2) Compared with naphthalene-derived mesophase pitch, the viscosity of the prepared ethylene-tar-derived mesophase pitch is relatively high, and the infrared spectrum shows a more peri-condensed structure. More ether bonds have been introduced, but there are more methylene, naphthenic, and aliphatic hydrocarbons at the same time; thus, mesophase pitch with a 100% streamlined polarized structure and moderate viscosity can be obtained.

(3) The content and texture of the mesophase pitches have influence on the structure and thermal conductivity of the carbon fibers. Mesophase pitch with 100% mesophase content can be prepared using ethylene tar as a raw material and naphthalene as a catalyst, which is an excellent precursor for preparing pitch-based carbon fibers with high thermal conductivity.

## Figures and Tables

**Figure 1 polymers-16-00970-f001:**
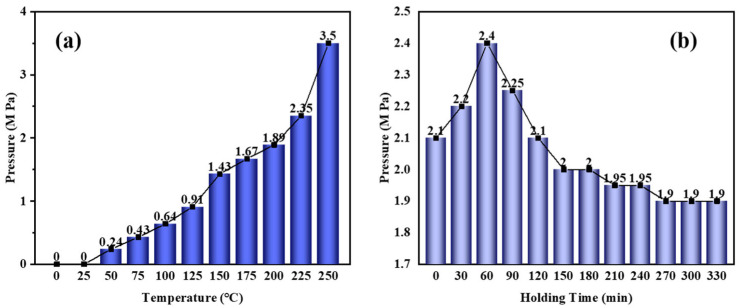
The curves of pressure versus (**a**) temperature and (**b**) holding time at 220 °C.

**Figure 2 polymers-16-00970-f002:**
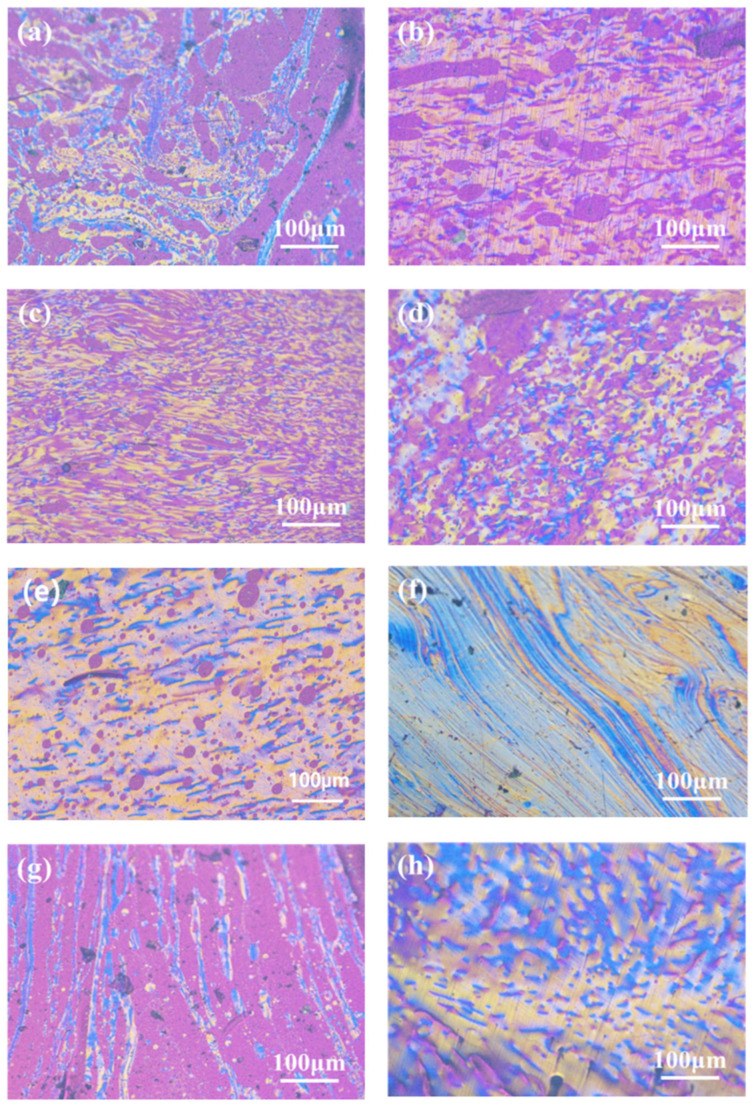
Polarized-light micrographs of (**a**) MP-20-210, (**b**) MP-25-210, (**c**) MP-30-210, (**d**) MP-20-220, (**e**) MP-25-220, (**f**) MP-30-220, (**g**) MP-0-220, and (**h**) MP-N.

**Figure 3 polymers-16-00970-f003:**
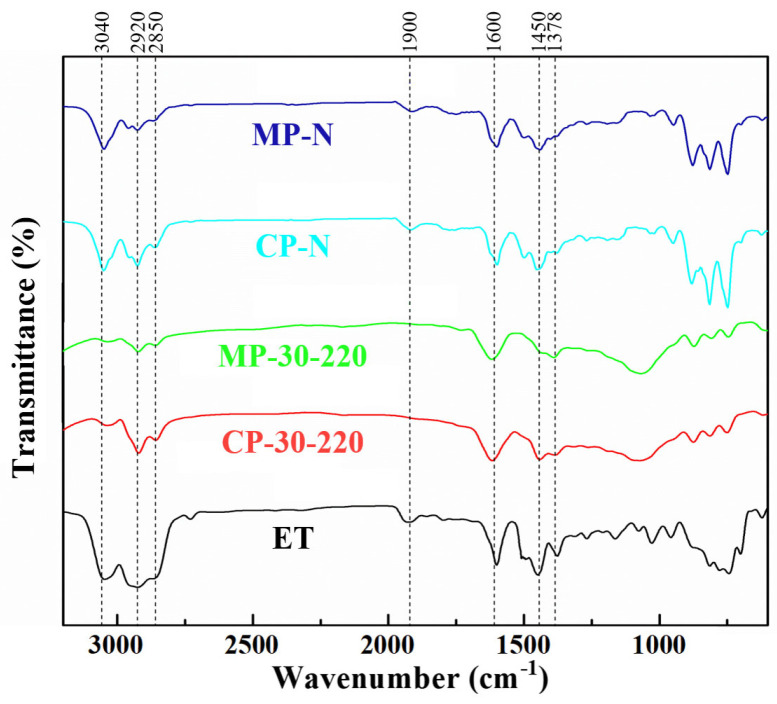
FT−IR spectra of ET and prepared pitches.

**Figure 4 polymers-16-00970-f004:**
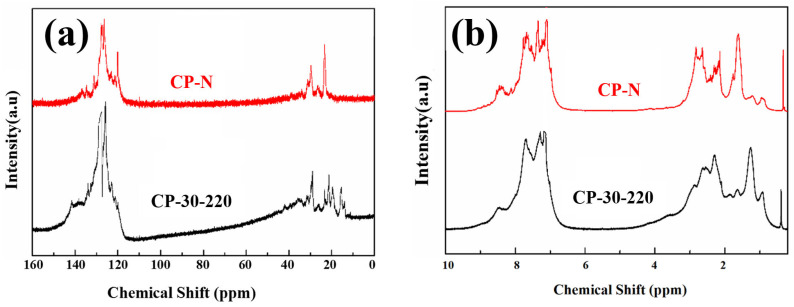
(**a**) ^13^C-NMR and (**b**) ^1^H-NMR of CP-30-220 and CP-N.

**Figure 5 polymers-16-00970-f005:**
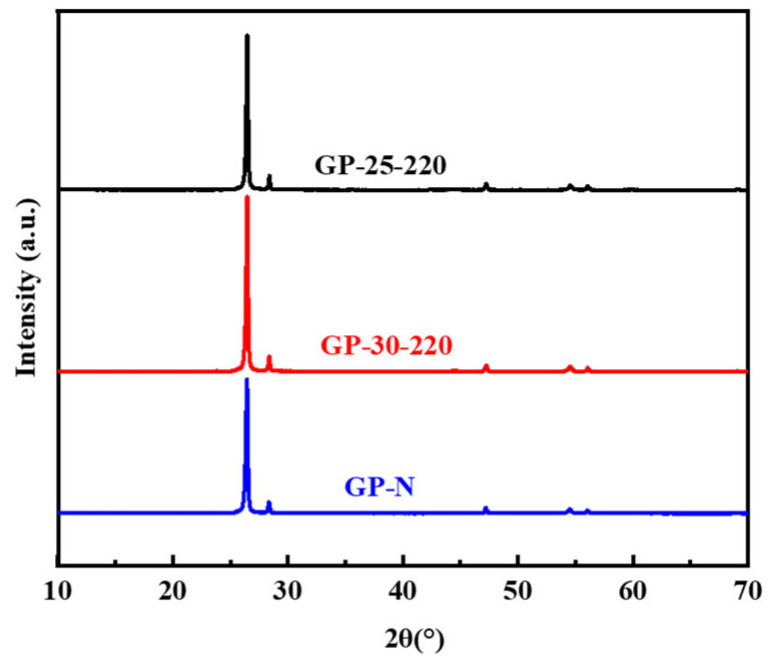
XRD of GP-25-220, GP-30-220, and GP-N.

**Figure 6 polymers-16-00970-f006:**
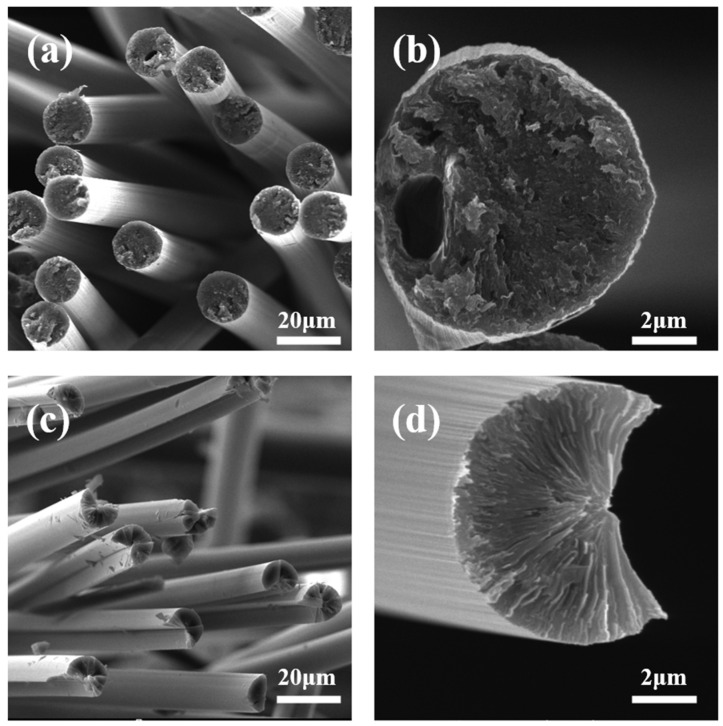
SEM images of (**a**,**b**) CF-25-220 and (**c**,**d**) CF-30-220.

**Table 1 polymers-16-00970-t001:** Effect of mass fraction of naphthalene on catalytic reaction pressure and yield.

Mass Fraction of Naphthalene (%)	Pressure (MPa)	Yield (%)	Experimental Phenomena
0	>6.00	—	Bursting disc damage
5	>6.00	—	
10	2.80	78.6	
15	2.75	80.8	
20	2.25	85.6	
25	2.40	93.7	
30	2.30	79.1	
35	3.00	56.2	
40	3.20	22.8	
45	3.20	—	Only oily products are obtained
50	3.10	—	

**Table 2 polymers-16-00970-t002:** Average structural parameters of CP-N and CP-30-220.

Pitch	H_ar_[a]	H_α_[b]	H_β_[c]	H_γ_[d]	H_ar_/H_al_	C_ar_[e]	C_al_[f]	C_CH2_[g]
CP-N	50.1	27.4	17.8	4.7	1.0	74.3	25.7	16.40
CP-30-220	48.3	30.9	15.9	4.9	0.93	70.8	29.2	18.21

[a] Hydrogen attached to aromatic carbons, 9.5~6.3 ppm; [b] hydrogen attached to aliphatic carbons at α-position to an aromatic ring, 5.2~2.1 ppm; [c] hydrogen attached to aliphatic carbons at β-position to an aromatic ring, 2.1~1.1 ppm; [d] hydrogen attached to aliphatic carbons at γ-position to an aromatic ring, 1.1~0.5 ppm; [e] aromatic carbon, 150.5~100.5 ppm; [f] aliphatic carbon, 50.5~0 ppm; [g] methylene carbon, 50.5~25 ppm.

**Table 3 polymers-16-00970-t003:** The crystal parameters of the graphite from mesophase pitch treated at 3000 °C.

Pitch	2θ (°)	d_002_	La (nm)	Lc (nm)	G (%)
GP-25-220	26.4974	0.3361	134.6104	43.3917	91.79
GP-30-220	26.5104	0.3360	174.6737	48.6194	93.68
GP-N	26.5124	0.3359	183.0504	52.4066	93.97

**Table 4 polymers-16-00970-t004:** The spinning parameters of MP-25-220 and MP-30-220.

Pitch	Breakage Frequency (time)	Continuous Spinning Time (min)	Minimum Diameter (μm)
MP-25-220	8	<10	20~25
MP-30-220	0	>40	13~16

**Table 5 polymers-16-00970-t005:** The tensile strength, tensile modulus, and thermal conductivity of CF-25-220 and CF-30-220.

Pitch	Tensile Strength (GPa)	Tensile Modulus (GPa)	Thermal Conductivity (W (m·K)^−1^)
CF-25-220	2.2	620	410
CF-30-220	2.6	850	730

## Data Availability

Data supporting the results of this study are available from the corresponding author.

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
