# Peer review of "Preparation of Mesophase Pitch with Fine-Flow Texture from Ethylene Tar/Naphthalene by Catalytic Synthesis for High-Thermal-Conductivity Carbon Fibers"

_polymers, 2024, doi:10.3390/polym16070970_

Round 1

Reviewer 1 Report

Comments and Suggestions for Authors

In this paper, mesophase pitch was synthesized using ethylene tar and naphthalene with boron trifluoride diethyl etherate as a catalyst, and carbon fibers were produced. This paper is very interesting and meaningful for the production of high-performance carbon fibers. I recommend that this paper is published after minor revisions according to a following comments.

Comments:

Lines 83-89

Although there is an explanation for the abbreviations CP-X-T and MP-X-T, there is no explanation for the abbreviations GP-X-T and CF-X-T. The authors should explain in the corresponding part of the “Materials and Methods” section.

Lines 186-189

Can the sample “MP-0-220” be synthesized? The yield of MP-0-220 is indicated as "-" in Table 1.

Line 189

without catalyst --> without naphthalene

Lines 209-211

The peaks at 3040 cm-1 (aromatic CH stretching vibration) and 750-870 cm-1 of naphthalene-derived catalytic polymerization pitch and mesophase pitch show stronger signals, --> The peaks at 3040 cm-1 (aromatic CH stretching vibration) and 750-870 cm-1 of CP-N and MP-N show stronger signals those of CP-30-220 and MP-30-220, 

L247

Figure 4 --> Figure 5

Figure 5

There are two “CP-25-220” and no “CP-30-220” in the figure.

Line 281

GF-30-220 --> CF-30-220

Line 282

GF-25-220 --> CF-25-220

Line 288

Figure 6. SEM images of CF-25-220 and CF-30-220. --> Figure 6. SEM images of (a, b) CF-25-220 and (c, d) CF-30-220.

Liners 289-290

GF-25-220 and GF-30-220 -->CF-25-220 and CF-30-220

Table 5

GP-25-220 --> CF-25-220

GP-30-220 --> CF-30-220

Reviewer 2 Report

Comments and Suggestions for Authors

·      The study presents a novel approach for synthesizing mesophase pitch using ethylene tar as a raw material and boron trifluoride diethyl etherate as a catalyst. While the research addresses the limitations of previous methods by reducing toxicity and causticity, and introducing a reaction-controlled technology, there are areas for improvement also.

·      The study lacks a comprehensive comparison with existing methods for mesophase pitch synthesis, particularly those utilizing alternative catalysts or raw materials. Without such comparisons, the significance and advantages of the proposed method remain unclear.

·       Although the use of boron trifluoride diethyl etherate reduces toxicity compared to HF/BF3 catalysts, a detailed toxicity assessment study will address potential environmental hazards associated with the new catalyst.

·      The study should provide thorough exploration of the optimization of reaction conditions, such as catalyst concentration, temperature, and reaction time.

·      The study investigates the structure and properties of the synthesized mesophase pitch and resulting carbon fibers, it needs tests, such as mechanical strength, thermal stability, and conductivity, would provide a more holistic assessment of the synthesized materials. I have following queries which are to be answered;

1.       How does the study ensure the complete removal of boron trifluoride diethyl etherate during the synthesis process to prevent residual toxicity in the final product?

2.       Were optimization studies conducted to determine the optimal reaction conditions for mesophase pitch synthesis using ethylene tar and boron trifluoride diethyl etherate?

3.       What additional performance tests were conducted on the synthesized mesophase pitch and carbon fibers to evaluate their suitability for practical applications?

4.       Has the study considered the potential environmental impact of the proposed synthesis method, particularly regarding the disposal of the catalyst and any by-products generated during the process?

5.       What efforts were made to optimize reaction conditions such as catalyst concentration, temperature, and reaction time to maximize the efficiency of the synthesis process?

6.       How does the study ensure the complete removal of residual catalysts and by-products to minimize environmental impact during the synthesis process and disposal?

7.       Could the study conduct additional performance tests on the synthesized mesophase pitch and carbon fibers, such as mechanical strength, thermal stability, and conductivity, for a more comprehensive evaluation?

Comments on the Quality of English Language

average
